

Comparison of high frequency, in-situ water quality analysers and sensors with conventional
water sample collection and laboratory analyses: phosphorus and nitrogen species
Steven J. Granger[1], Juan A. Qunicke[12], Paul Harris[1], Adrian L. Collins[1], Martin S. Blackwell[1]
[1]Sustainable Agriculture Sciences, Rothamsted Research, North Wyke, Okehampton, Devon,
EX20 2SB, U.K.
[2]Instituto National de Investigación Agropecuaria, La Estanzuela, Ruta 50, Km. 11, Colonia,
Uruguay.
Correspondence to: S. J. Granger (steve.granger@rothamsted.ac.uk)
Abstract
The long-term collection of water samples for water quality analysis with high precision
laboratory instrumentation is routine in monitoring programmes however, such sampling is
labour intensive, expensive, and therefore undertaken at a low temporal frequency. Advances
in environmental monitoring technology however, mean that it is now possible to collect high
temporal frequency measurements for a wide range of water quality parameters without the
need for the physical collection of a sample. The downside to this approach is that the data can
be subject to more 'noise', due to environmental and instrument variables. This raises the
question of whether high frequency, lower precision data are better than low frequency, higher
precision data. This study reports the findings of an investigation of agricultural land drainage
comparing measurements of total phosphorus (TP), total reactive phosphorus (TRP),
ammonium ($NH_4$-N) and total oxidised inorganic nitrogen (NOx-N) collected using both
equipment in situ and concurrent water samples analysed in the laboratory. Results show that
both in situ PHOSPHAX TP and NITRATAX NOx-N instruments can provide comparable





data to that measured using samples analysed in the laboratory; however, at high discharge and
low NOx-N concentrations, the NITRATAX can under report the concentration. In contrast,
PHOSPHAX TRP and YSI sonde $NH_4$-N data were both found to be incomparable to the
laboratory data. This was because concentrations of both parameters were well below the
instruments accurately determinable level, and because the laboratory data at low
concentrations were noisy.
Keywords: water quality; phosphorus, nitrogen, ammonium, sensors; in situ; runoff; field
drainage
1.  Introduction
Long-term routine, but infrequent, water quality sampling used widely in strategic scale
monitoring provides insight into longer-term trends (Howden et al., 2010). However, such
sampling fails to capture higher resolution data necessary for insight into hydrological and
biogeochemical processes and responses (Granger et al., 2010) including evidence of non-
stationarity, self-organisation, and fractals (Harris and Heathwaite, 2005; Milne et al., 2009;
Kirchner and Neal, 2013). Advances in environmental monitoring technology mean that it is
now possible to collect high resolution measurements of a wide range of water quality
parameters, providing detailed insight into hydrochemical temporal dynamics. Technologies
vary depending on the parameters being measured, but typically include, automated wet
chemistry apparatus in situ (e.g. for phosphorus (P) analysis) or ultra-violet optical sensors (e.g.
for total oxidised nitrogen) (Palmer-Felgate et al., 2008; Donn et al., 2012; Carey et al., 2014;
Skeffington et al., 2015; Bieroza and Heathwaite, 2015; Mellander et al., 2016). Frequency of
measurements vary, ranging from every minute (or less) to hourly, depending upon the
parameter, but are more typically undertaken at 15-minute intervals. Wet chemistry in situ



analysers and optical sensors have been shown to deliver important insights into nutrient
fraction dynamics in response to runoff (Mellander et al., 2015) and catchment management
(Perks et al., 2015). High resolution sampling and analysis in situ captures a broader range of
pollutant concentrations than routine infrequent sampling and thereby elucidates hysteresis,
diurnal patterns and non-storm dependent transfers (Heffernan and Cohen, 2010; Bende-Michl
et al., 2013). Monitoring in situ can be used to identify pollutant transfer typologies. For
example, Jordan et al. (2005) used in situ wet chemistry analysis to detect three types of total
P (TP) transfer events: chronic storm-independent transfers reflecting on-farm slurry and
fertiliser applications; acute storm-dependent transfers associated with agricultural diffuse
pollution, and; acute storm-independent transfers reflecting specific incidental pollution
events. In situ devices remove sample storage requirements and provide a means of avoiding
water sample storage-associated chemical transformations (Bende-Michl and Hairsine, 2010).
Previously, studies were limited to the collection of water samples either manually or using
automated water samplers, and then transfer of samples to laboratories for analysis by wet
chemistry and colourimetric methods. However, despite transforming the hydrologic sciences
over the past 50 years (Rode et al., 2016), questions remain about the precision of
measurements made using these technologies relative to standard sample collection and
laboratory analysis. The traditional auto-sampler approach followed by laboratory analyses of
nutrient content can carry risks and uncertainties associated with a number of problems,
including small sampling volume, preferential sampling effects, limited coverage of the stream
cross-section and transformation risks during storage in conjunction with time delays between
sample collection and subsequent laboratory analyses (Kotlash and Chessman, 1998; Harmel
et al., 2006; McMillan et al., 2012). Storage-associated transformations are caused by a range
of physical and biochemical processes including hydrolysis, sorption, precipitation, microbial
uptake or release and complexation (Jarvie et al., 2002a; Harmel et al., 2006). Previous work



(McMillan et al., 2012) has suggested that biogeochemical effects during sample storage can
contribute more to uncertainty than errors due to preferential sampling or lower extraction of
sediment-bound nutrients. The greatest proportional losses of dissolved nutrients in stored
water samples occur when concentrations are low, with losses up to 50 % for nitrate and 67 %
for soluble reactive P after six days of storage with no refrigeration (Kotlash and Chessman,
1998). Such uncertainties are also reported by Lloyd et al. (2016) who describe an almost 10-
fold increase in uncertainty of both nitrate and TP loads measured over 2 years in a river in the
U.K. when comparing laboratory and automated sensor data.
This raises the question of whether high frequency, low precision data is better than low
frequency, high precision data. Rode et al. (2016) recognise that there are major issues related
to calibration of automated sensing equipment and the need for regular servicing, along with
a pressing need for the development of automated tools and standards for data quality
assurance (Campbell et al., 2013). There is still much work to be done in quantifying the
precision of automated water quality sensors, and accordingly, herein we report the findings
of a study comparing measurements of TP, total reactive P (TRP), ammonium nitrogen ($NH_4$-
N) and total oxidised inorganic nitrogen ($NO_x$-N) collected using both automated equipment
and concurrent water samples analysed in the laboratory. We test the hypothesis that in situ
measurements of these parameters can be as precise as those acquired by laboratory analysis
of manually collected water samples analysed using standard laboratory techniques.

2.  Material and Methods
2.1 Study Site
The study was undertaken on the North Wyke Farm Platform (NWFP), an instrumented
research grassland farm of 63 ha, split into 15 hydrologically isolated sub-catchments, over



which three different 21 ha livestock and grassland management systems are imposed (Orr et
al., 2016). From April 2013 to July 2015, all 15 NWFP sub-catchments were assigned to one
of three treatments: (i) permanent pasture ('green' farmlet); (ii) increased use of legumes
(blue farmlet), and; (iii) innovation (red farmlet), via a gradual and planned re-seeding
campaign (Figure 1a). The soils of the NWFP are clay loams (Figure 1b). Within each sub-
catchment a range of instrumentation takes measurements on water, air and soil parameters in
situ, much of this data being at a high temporal resolution (15 mins).
2.2 Sampling Strategy
For this study, one sub-catchment from within each of the three management systems was
chosen for investigation (numbered 2, 5 and 8; Figure 1b) of drainage caused by a rainfall
event on the 3rd December 2015. A tipping bucket rain gauge (Adcon, Austria) located in the
centre of each catchment measured the rainfall at a resolution of 0.2 mm per tip. During this
event, measurements of discharge, TP, TRP, $NH_4$-N and $NO_x$-N were taken using the
instrumentation in situ. Auto-samplers (Teledyne ISCO, New England, USA) were used to
sample the discharge automatically at pre-determined flow thresholds. Manual grab samples
were also collected throughout the discharge event and both these and the auto-sampler
samples were analysed in the laboratory. Grab samples were taken to both sample the
discharge before and after the main storm drainage, and a sub-set during the storm drainage
were taken at exactly the same time as the automated in situ analysis so as to generate paired
results for TP and TRP. Grab samples were kept cool, and a sub-sample filtered through a
0.45 µm cellulose nitrate filter, before all samples were analysed in the laboratory within 48
hours.
2.3 Measurement of Discharge


Each of the sub-catchments drain, via French drains, to a monitoring station where H flumes
are located with a capacity designed for a 1 in 50-year storm event. The flumes intercept and
channel drainage in such a way that discharge can be determined by a rating curve calculated
based on the the height of the liquid in the flume. Drainage in this context is defined as all the
water that moves from the sub-catchment through the flume irrespective of its hydrological
pathway. Water heights within the flumes were measured by pressure level sensors (OTT
hydromet, Loveland, CO., USA). These sensors measure the depth of water by means of an
integrated controller and ceramic pressure-measuring cell. The level offset (to the flume bed)
was checked fortnightly and updated, if required, in the logger software.
2.4 Automated in situ Measurements
2.4.1    Phosphorus
Total P and TRP are measured in a sample collected from a sump at the monitoring station by
a separate device (SIGMATAX 2, Hach, Salford, UK) which homogenises an unfiltered
sample using ultra-sound before passing it to a process photometer (PHOSPHAX sigma,
Loveland, Colorado, USA). The analyser analyses ortho-phosphate colourimetrically using
standard molybdenum blue chemistry. Total P samples are digested prior to colourimetric
analysis by heating, under pressure, with sulphuric acid and sodium peroxydisulphate while
TRP analysis occurs on an undigested sample. The PHOSPHAX was calibrated daily through
the running of an internal standard; however, it was not possible to run further quality
controls or references. The lowest concentration the instrument is reported to measure is 50
($\pm$ 1) $\mu$g $PO_4$-P $l^{-1}$.
2.4.2    Nitrogen
Drainage the sump in the conduit upstream of the flume is automatically pumped every 15
mins into a purpose built stainless steel by-pass flow cell that houses the sensors. Water is





pumped into, and out of, the base of the flow cell and this, coupled with the V shape design,
ensures that there is no retention of sediment or particulate matter either between samples or
over time. Within the flow cell, $NH_4$-N ($NH_4$ + ammonia) is measured using an ion selective
electrode contained within a multi-parameter sonde (6600V2, YSI, Hampshire, UK). Total
oxidised inorganic nitrogen is measured by a self-cleaning, optical UV absorption sensor
(NITRATAX Plus SC, Loveland, Colorado, USA). There is no specified lowest working
concentration for this sonde; however, as it has an accuracy of $\pm$ 2 mg $NH_4$-N $l^{-1}$ at its lower
range, anything below this is considered 'not accurate' (YSI, Ohio, USA. *pers. comm.*).
Oxidised inorganic nitrogen dissolved in water absorbs UV light at wavelengths below 250
nm, so by passing UV light through the medium and measuring the absorption using a 2-
beam turbidity compensated photometer, the NOx-N concentration is calculated. The lowest
accurately determinable concentration for the instrument is $0.503 \pm 0.5$ mg NOx-N $l^{-1}$.
Both probes were calibrated monthly and drift corrected, but no additional in situ quality
controls were applied.
2.5 Laboratory Measurements
Unfiltered samples presented to the laboratory were analysed for both TP and RP thus giving
equivalent data to that generated from the Phosphax instruments (i.e. TP and TRP).
Samples requiring TP analysis were initially subject to an oxidation reaction using acidified
potassium persulphate thus converting all P forms to RP. Both digested and undigested
samples were then analysed for RP colourimetrically on an Aquachem 250 analyser using a
molybdenum blue reaction (Murphy and Riley, 1962). The limits of quantification (LOQ: the
lowest accurately determinable concentration) for TP and RP were 10 ($\pm$ 1.4) and 2 ($\pm$ 0.04)
µg $PO_4$-P $l^{-1}$, respectively. The accuracy of TP digestions was checked using quality controls
which were always within 8 % of the target value and with 78 % being within 5 %. Similarly,



quality controls were run during the analysis of RP which were always within 5 % of the
target value.
Unfiltered samples were also analysed colourimetrically for $NH_4$-N and $NO_x$-N on the
Aquachem 250 analyser. Total oxidised inorganic nitrogen was determined through the
reduction of nitrate to nitrite by hydrazine sulphate and total nitrite is diazotized with
sulphanilamide and coupled with N-1-naphthylethylenediamine dihydrochloride to form an
azo dye with an absorbance maximum at 540 nm. The LOQ for this method was 0.1 (± 0.003)
mg $NO_x$-N $l^{-1}$ and quality controls were always within 3 % of their target.
Ammonia/ammonium was determined by the chlorination of ammonia with sodium
dichloroisocyanurate to monochloramine, which reacts with salicylate to form a second
intermediate, 5-aminosalicylate.  Oxidative coupling of 5-aminosalicylate with salicylate
forms an indophenol dye with an absorbance maximum at 660 nm.  Nitroprusside stabilises
the monochloramine intermediate and also promotes the final oxidative coupling stage. The
LOQ for this method was 0.4 (± 0.01) mg $NH_4$-N $l^{-1}$ and quality controls were always within
5 % of their target.
2.6 Data Pre-processing
For TP and TRP, the manual grab sampling and in situ flume measurements only occurred
simultaneously on five out of 30 occasions for all three sub-catchments, thus only five paired
samples could be compared directly, with the same time stamp. For the nitrogen species,
measurements only occurred simultaneously on one of three occasions. Thus, to make efficient
use of all the grab sampling data, the in situ flume chemistry data were infilled (or predicted)
to provide an exact match to the grab sampling times. This was achieved using a splines fit (via
the na.spline() function in the 'zoo' R package of Zeileis and Grothendieck (2005)). Outputs
of prediction uncertainty for the infilled data were not sought, although future work could





transfer this uncertainty into the subsequent relationship analyses (e.g. via weighted correlation
or regression analysis). In this respect, all infilled in situ data points are effectively viewed as
measured in situ data for subsequent statistical analyses (this assumption is still checked
visually, however).  Constraints were also set in place to ensure the infilling did not provide
values below zero or provide values at a higher level of precision than that measured.

197        2.7 Statistical Procedures

Once the infilling had been conducted, paired in situ flume and laboratory grab sampling data
were graphically related using time series and scatterplots for all four water quality parameters.
Time series plots are useful in that they can indicate systematic effects, such as sustained
periods of over- or under-estimation, but where the general temporal pattern of the data is
retained. The time series plots also provide an important visual assessment of the spline
infilling procedure described above. For scatterplots, if the two methods of measurement were
an exact match, then they should lie on the 45º line. Data points that lie below the 45º line
indicate where the in situ data under-estimates the laboratory data, and vice-versa.
These visual summaries were complemented by a basic set of statistical goodness-of-fit
diagnostics. The intercept and slope parameters from linear regression fits (between the in situ
and laboratory data) are found, together with $p$-values for significance from zero and from one,
respectively. Associated $R^2$ values from the same regressions are also reported and should tend
to 1. Mean error (ME), root mean squared error (RMSE) and Normalised RMSE (NRMSE)
values are reported (via functions in the 'hydroGOF' R package), where all three diagnostics
should tend to zero. In this case, the *errors* referred to *in situ minus laboratory* data, thus a
negative ME value would indicate that the in situ data under-estimates the laboratory data, on
average. RMSE reflects the variance of the errors, which ideally needs to be as small as
possible. The NRMSE diagnostic is a relative measure of RMSE, and thus relays quite clearly



when the in situ data has a good or poor correspondence with the laboratory data, regardless of
different scales of measurement from the different sub-catchments.
A final, but limited analysis was also conducted on the genuine paired samples found for TP
and TRP only - i.e. only five pairs for each sub-catchment. This data was analysed using paired
*t*-tests and analysis of variance (ANOVA), and was presented using Tukey mean-difference
plots. Further analyses could have considered random sampling for five pairs from the infilled
data of 30 pairs, and repeating the tests considered here, on each random sample. This would
assess the sensitivity of the results to sample variation and to an extent, the infilling. However,
this was considered beyond the scope of this study; and in any case, the outcomes would always
be severely limited due to the very small sampling size.
All statistical analyses were conducted in R (https://www.r-project.org/), where in all cases,
the in situ data were compared to the *unfiltered* laboratory data.

3.  Results
3.1 Data summaries
In the first instance, it is useful to summarise the measured data, where infilled data or paired
data are not needed. In this respect, sample size and the ranges (minimums to maximums) for
TP, TRP, $NO_x$-N and $NH_4$-N measured in the drainage from sub-catchments 2, 5 and 8,
obtained by both the automated in situ analysers and laboratory analysed manually collected
samples are presented in Table 1. Values of TP ranged between 40 to 770 µg P l$^{-1}$ and for TRP
between 0 to 70 µg P l$^{-1}$ as measured by the in situ Phosphax analysers. For $NO_x$-N and $NH_4$-
N, the values measured in situ ranged between 0.62 to 4.8 mg N l$^{-1}$ and 0.04 to 1.5 mg N l$^{-1}$,
respectively. The range of values measured in the manually collected samples analysed in the
laboratory, in general, compare favourably to the in situ data. This is even though there are





much fewer data and that potential highs and lows in concentration could have been missed in
the manually collected samples.

242       3.2 Chemistry Response to Discharge

Data on the discharge from the three sub-catchments and rainfall is presented in Figure 2. The
results show three similar twin peaked hydrographs but with different magnitudes of peak
discharge of 15, 22 and 28 l s$^{-1}$ for sub-catchments 2, 5 and 8, respectively. The different scales
of the hydrographs reflect differences in, amongst other things, sub-catchment size, rainfall,
slope, soil moisture and soil type. In all three sub-catchments, TP data from both in situ
analysers and the laboratory analysed grab samples exhibited a positive relationship with
discharge (Figure 3a-c). The highest values of TP were associated with the initial smaller peak
in discharge, and a latter smaller peak in TP associated with the second, large, peak in
discharge. In all cases, the chemographs generated by both analytical approaches appear similar
and match the responses reported elsewhere (Heathwaite and Dils, 2000; Granger et al., 2010;
Lloyd et al., 2016). Such relationships with discharge are less clear with the lower
concentration TRP data (Figure 3d-f). In situ TRP concentration data again exhibit a positive
relationship with discharge, and possibly even a two peaked chemograph, similar to that of the
TP data. However, the low concentration range compared to that of the TP, means that when
the data is rounded to the nearest 10 µg P l$^{-1}$, the resolution of the chemograph is severely
affected and detail is lost. The TRP data generated via laboratory analysis are not subject to
this rounding effect; however, these data exhibit considerably more 'noise', and while it is
possible to visualise some relationships with discharge, in all but the data from sub-catchment
8, this is highly subjective.
The NOx-N chemographs generated by the in situ analysers and the laboratory analysed
samples display the classic dilution effect reported elsewhere (Webb and Walling, 1985;
Granger et al., 2010; Lloyd et al., 2016) with concentrations dropping rapidly with the onset of





increased discharge, and slowly recovering to pre-storm flow values over time on the falling
limb of the hydrograph (Figure 4e-f). The data generated for $NH_4$-N from the in situ sensors
clearly show a positive relationship with discharge from all sub-catchments and, interestingly,
even a second $NH_4$-N peak on the chemograph of sub-catchment 8 associated with the main
spike in discharge (Figure 4a-c). This positive relationship is not unusual (House and Warwick,
1998a; Inamdar, 2007; Fucik et al., 2012), although it tends to be much lower in concentration
compared to $NO_x$-N and often this is not very discernible as the $NH_4$-N is rapidly nitrified to
$NO_x$-N (House and Warwick, 1998b). Where high concentrations of $NH_4$-N occur as spikes
associated with discharge, it is often more related to incidental losses of recently applied $NH_4$-
N as a result of farmland management practices (Granger et al., 2010). Data generated from
the laboratory analysed grab samples provide a slightly more mixed picture. Where
concentrations were highest (sub-catchment 8), these data appear to confirm the positive
relationship of $NH_4$-N with discharge, even reproducing the second $NH_4$-N peak. In sub-
catchment 5, where $NH_4$-N concentrations were lowest, the laboratory data are noisier, but it
is still possible to observe an increase in $NH_4$-N concentration with increased discharge. In sub-
catchment 2, however, the laboratory $NH_4$-N data show no relationship with discharge (Figure
4a).
In all chemographs (Figures 3-4), the outcomes of the in situ spline infilling described above,
is shown. Here in filling never required a difficult extrapolation, but instead was always a
simple interpolation that was richly informed by actual measured data that were temporally
similar. Clearly, no unusual predictions result and the infilling should be considered reliable,
and can be safely viewed as strongly comparable to the in situ data for subsequent statistical
analyses.

3.3 Comparison of In Situ and Laboratory Analysis





The data obtained for genuine paired laboratory analysed manual grab samples and
PHOSPHAX in situ TP and TRP are presented in Table 4 (it is of no value to do this for nitrogen
species, as only one to three genuine pairs were available). The differences between the two
sets of data are reported relative to the laboratory data which have been subject to full analytical
quality control. Using this comparison, it can be seen that differences between TP values are
lower than for TRP, with respective ranges between +56 to -30 µg P l$^{-1}$ (+29 % to -38 %) and
+13 to -33 µg P l$^{-1}$ (+186 % to -57 %).
The difference between the two methods of measurement were assessed using paired $t$-tests.
The average difference between laboratory and in situ values for TP was -3.933 (standard error
of difference 4.947, 95 % CI -14.54, 6.677) and the standard deviation of differences was 19.16.
The $t$-test for TP indicated that there was no evidence of a difference between laboratory and
in situ measurements ($t_{14}$ = -0.8, p = 0.44). However, the average difference between lab and
in situ values for TRP was 8.933 (standard error of difference 3.534, 95 % CI 1.353, 16.51)
with the standard deviation of differences being 13.69. This indicated that for TRP, that there
was evidence of a statistically significant difference between laboratory and in situ
measurements ($t_{14}$ = 2.53, p = 0.024).
Differences between the two measurement methods was also assessed using ANOVA in order
to take into account that the data came from three different sub-catchments. This did not suggest
any influence of the sub-catchment difference on the size of measurement difference. Tukey
mean-difference plots are presented in Figure 5 and plot the difference between the two values
against the average of the two measured values. Limits of agreement (dashed lines) are plotted
at ± 2 standard deviations from the mean difference and indicate the range that approximately
95 % of the data is expected to fall in. From these plots the data suggest that, while there is no
obvious trend in TRP data, differences in the TP values are greater at lower concentrations with



the laboratory generating higher values but that this difference reduces as the TP concentration
increases.
3.4 Comparison of Modelled In Situ and Laboratory Analysis
Given the small sample number of actual in situ and laboratory analysed grab samples,
assessing the differences between these two approaches is extremely limited. We therefore
compare the modelled in situ and laboratory analysed samples. This is because we consider
error in the data obtained from the laboratory to be relatively low (Madrid and Zayas, 2007),
with these data subject to analytical quality controls and checks. Any difference between in
situ values and the laboratory must therefore be explained via other processes and mechanisms.
3.4.1    Phosphorus
For TP and TRP, the resultant paired data is presented using scatterplots in Figure 6. In all
cases, the ideal 45º line is shown together with the actual linear fit. Results of the tests for
whether or not the ideal and actual lines significantly deviate from each other are given in Table
2, together with a general fit measure in $R^2$. At the 95 % level of significance, only the
laboratory and in situ data for TP in sub-catchments 5 and 8 are in good agreement (as the *p*-
values in Table 2 indicate the intercepts and slopes of their fitted lines are not significantly
different to zero or one, respectively). Laboratory and in situ TP data in all three sub-
catchments do however provide relatively high $R^2$ values, where for sub-catchment 2, the in
situ TP tends to under-estimate laboratory TP at high values, pivoting the fitted line downwards
at these values. Table 3 provides the ME, RMSE and NRMSE results for TP and TRP, where
the negative ME value for TP in sub- catchment 2, indicates an overall under-estimation of
laboratory TP by in situ TP, whilst in the other two sub-catchments, the reverse is true.
Although sub-catchment 2 does not indicate the strongest 1:1 relationship between the paired





TP data, its TP data are most alike in terms of variation - as seen by the least scatter around the
fitted line, coupled with the lowest NRMSE value.
Corresponding results for TRP are not promising (Figure 5, Tables 2 and 3), where each
scatterplot depicts a poor correspondence between the laboratory and in situ TRP data, and
these poor relationships are statistically endorsed by the test results and the low $R^2$ values
presented in Table 2. Diagnostics presented in Table 3, provide little further insight into the
behaviour of the paired TRP data, except that in situ TRP will tend to under-estimate laboratory
TRP (as MEs are negative in two sub-catchments). Note however, that in situ TRP tends to be
less variable than laboratory TP, as shown by the scatterplots.

346        3.4.2    Nitrogen

Results for the differences between the in situ and laboratory $NH_4$-N data are quite complex.
From the scatterplots in Figure 7, and the associated tests in Table 2, a 1:1 relationship between
the paired $NH_4$-N data in sub-catchments 2 and 8 is clearly absent, although within sub-
catchment 2 the data are influenced by an anomalously high laboratory $NH_4$-N result. However,
the paired $NH_4$-N data do provide a high $R^2$ value in catchment 8, indicating a reproducible
relationship of sorts, albeit not one that is ideal. The most promising relationship for the paired
$NH_4$-N data is found in sub-catchment 5, where the $R^2$ value is reasonable and the NRMSE
value is much lower than that found in the other two sub-catchments.
Results for the differences between the paired NOx-N data, in contrast, are quite promising.
The scatterplots in Figure 7, overall, show a reasonable correspondence between the paired
NOx-N data, for all three sub-catchments, which is endorsed by very high $R^2$ values in Table
2. Although in all cases, these relationships cannot be viewed as 1:1 as indicated by the test
results in Table 2. For all cases, the in situ NOx-N data tends to over-predict the laboratory
NOx-N data.




4. Discussion
4.1 Phosphorus
Direct comparison of laboratory and the in situ TP data shows no evidence of a significant
difference although, at lower concentrations, the in situ data would appear to be *lower* than the
laboratory values. Here, however, it is important to bear in mind that no direct comparisons
were made on samples that were taken on discharge at the higher end of the concentration
range. The modelled data confirm that there is a good match, in general, between in situ data
and laboratory values with fitted lines not being significantly different to zero or one in sub-
catchments 5 and 8. In sub-catchment 2, conversely, it would appear that in situ data were
lower than laboratory values at *higher* concentrations which is confirmed by the negative ME
value for this sub-catchment. Irrespective of this, all data showed good correlations with
relatively high $R^2$ values, a fact that is confirmed by the good agreement shown by the
chemographs in Figure 3 a-c. The data suggest that for TP, the PHOSPHAX in situ analysis
provides reasonably good agreement with manual sampled laboratory analysed samples, and
conversely that the manual samples do not suffer excessively from systematic, sampling or
storage errors. However, it is noteworthy that the PHOSPHAX in situ data does produce a
relatively 'smooth' chemograph which is in contrast to the laboratory data which is noticeably
more 'noisy' and even contains a few anomalously high concentrations ('outliers') i.e. Figure
3a. This is probably a result of one or a combination of, three important issues regarding TP:
a) data generated in situ is 'smoothed' by the analyser by rounding values to the nearest 10 µg
P l$^{-1}$, b) sample container contamination at either the sampling stage or latterly during
laboratory digestion, and c) laboratory analytical error. In the first case, the in situ values might
actually be noisy, but this is not reflected in the smoothed data generated for download. In the
second case, it is assumed that P of unknown origin (biological, tap water, chemical) could





have randomly, as opposed to systematically, contaminated some equipment leading to a high
result. In this scenario it is hard to imagine how this sort of error could cause a lower result
than expected. In the third, it could just be a random analytical artefact, which has resulted in
an unusually high (but could also cause an unusually low) result.
In contrast, the comparison of the TRP data were far less conclusive. Direct sample pair
comparison indicated a significantly lower concentration measured in situ than that measured
in the laboratory. Further, the larger data set generated by comparing modelled in situ and
manually sampled laboratory analysed TRP data shows very poor correspondence with
significant differences in both slope and intercept being >0 and <1 in every case, respectively,
indicating that the in situ data were consistently lower than that measured in the laboratory.
The low $R^2$ further confirms poor replication of data, a fact further confirmed by the
chemographs presented in Figure 3 d-f. It can be seen from Figure 6 d-f that the main cause for
poor correlation between the two data-sets is probably down to a combination of two factors;
a) the low resolution of the PHOSPHAX in situ data, rounding all numbers to the nearest 10,
and b) more importantly, that the vast majority of the PHOSPHAX in situ data is lower than
the machine's analytical limit of 50 µg P l$^{-1}$. That being said, in situ TRP concentrations have
trends (Figure 3 d-f) which are not so clearly represented in laboratory TRP data which again
although being above LOQ are extremely noisy.
One explanation is sample degradation between sampling and analysis. Ideally, samples should
be analysed immediately after collection to minimise degradation effects, but sample storage
is usually unavoidable prior to analysis. The concentrations of dissolved nutrient within water
samples can vary during storage as the result of a wide range of physical, biological and
chemical processes including sorption, hydrolysis, precipitation, complexation, and microbial
uptake and release (Jarvie et al., 2002b). This is particularly relevant for the samples collected
in this instance since they were unfiltered prior to analysis such that they were of the same



matrix as the sample collected by the PHOSPHAX. Rapid filtration of samples (typically >0.45
μm) is usually recommended in order to exclude microbial cells and inorganic particulate
material, which can result in changes in physical or chemical forms of P through processes
such as microbial uptake or adsorption in unfiltered samples (Lambert et al., 1992; Jarvie et
al., 2002b; Worsfold et al., 2005). Biological processes and sorption to particulate matter can
be rapid; Lambert et al. (1992) reported that concentrations of 'dissolved' P decreased
substantially over a four-hour period in unfiltered lake water samples. Another possible effect
of the unfiltered matrix is that sample particulates could be causing noise in the laboratory TRP
analysis, both through their physical presence in the flow cell and through biogeochemical
alteration of the sample in reaction to analytical reagents (Jarvie et al., 2002b), although
presumably this is also an effect that happens in the PHOSPHAX analyser.
4.2 Nitrogen
Results for the differences between the in situ and laboratory $NH_4$-N data are quite complex
but the 1:1 relationship between the paired $NH_4$-N data were in general poor. However, the
paired $NH_4$-N data do provide reasonable $R^2$ values in sub-catchments 5 and 8. This variation
in the responses needs to be examined more carefully. From the chemographs in Figure 4 a-c,
all three in situ sondes produced similar responses, with rising and falling concentrations
matching rises and falls in discharge. This would seem to indicate that the sondes were
detecting a genuine chemical response. However, all the in situ data reported are well below
the accuracy of the sonde at this concentration (Figure 7 a-c). This could be as a result of other
factors affecting the sonde other than $NH_4$-N. The ion selective electrode is subject to effects
caused by changes in temperature and interference from ions, which are similar in nature to the
analyte. While the changes in temperature, or ions such as sodium and chloride, might only be
slight in response to field drainage, they could be enough to cause the small responses recorded
here which have a maximum range of about 1 mg $NH_4$-N $l^{-1}$ and which were always below the





recommended accurate working concentration for the instrument. That said, in sub-catchment
8 which had the highest recorded $NH_4$-N values both by the sonde and the laboratory, the
chemical response in $NH_4$-N *is* mirrored by the laboratory grab samples, giving the highest $R^2$
of 0.83. Laboratory concentration values were also in the main below LOQ, but were much
closer to analytical limits than that of the sonde, and at their peak, slightly exceeded it. This
would seem to indicate that the sonde response in sub-catchment 8 would appear to be genuine
even if the absolute $NH_4$-N concentration is suspect. If this is the case, then we can maybe
assume that the responses recorded in sub-catchments 2 and 5 are also genuine, even if their
absolute values may not be. Laboratory values from these two sub-catchments were, however,
well below LOQ so cannot be used to back up this conclusion. Interestingly, laboratory data
from sub-catchment 5 (which recorded the lowest $NH_4$-N concentration from any of the three
sub-catchments) does mirror the response of the sonde to a degree ($R^2 = 0.67$), while the values
from sub-catchment 2 show no similarity at all. This paradox is confusing as if the loss of
response from sub-catchment 2 was due to sampling and unfiltered storage losses because of
the low $NH_4$-N concentration (i.e. (Kotlash and Chessman, 1998; Lentz, 2013)) then that would
surely have occurred in the even lower concentrations of sub-catchment 5. Further storm period
analyses are required to help resolve this paradox.
The NOx-N data, in contrast, show good similarity, which although not significantly similar,
have a very high $R^2$ (Table 2) and in all cases, the in situ modelled NOx-N data tends to over-
predict the laboratory data. The reason for this is clear from the chemographs in Figure 4 d-f,
whereby the trends in both sets of data are virtually identical (leading to high $R^2$), but whereby
modelled in situ values and laboratory values differ at lower concentrations. In all three
examples, pre- and post-discharge event values are near identical, but with the onset of
increased discharge, the NOx-N concentrations drop, with laboratory values dropping further
than modelled in situ values. In all cases, none of the recorded values are below the instruments





working capabilities so should be considered reliable, and only a few values are below the
laboratory LOQ. The reason for this discrepancy is unclear and is the result, or a combination
of, either the sensor underestimating NOx-N at increased discharge, or the laboratory analysed
grab samples having a lower measured NOx-N at high discharge.
Here it useful to bear in mind that as the measurement is based on the evaluation of (invisible)
UV light, the colour of the medium has no effect. The sensor contains a two-beam absorption
photometer with turbidity compensation. So perhaps this turbidity compensation is having a
greater effect on reducing calculated NOx-N values in situ at times when turbidity is greatly
increased (at high discharge).

5. Conclusions
An increasing number of studies are reporting the use of in situ analysers and sensors to collect
high temporal resolution hydrochemical data. Whilst such data permit the use of exploratory
data interpretation techniques such as hysteretic loops, much hydrochemical data are used to
estimate time-variant or averaged concentrations in the context of environmental objectives or
thresholds and to estimate nutrient loads. The comparison herein of nutrient species data
collected using in situ analysers or sondes and manually collected laboratory analysed samples
confirms the following:
• PHOSPHAX in situ TP data would appear to be reliable, most likely as the
determined concentrations are nearly always more than the instrument's lower
limits. Discrepancies between laboratory and in situ data appear to increase as the
PHOSPHAX lower measurable limit is approached.
• PHOSPHAX TRP measurements, in the context of the field drainage described
here, are unreliable as the concentrations were nearly always below the LOQ for




the instrument. This is reflected in the poor agreement between laboratory and
instrument data. This poor agreement is largely due to the laboratory data being
very 'noisy' despite being above laboratory LOQ and may reflect
sampling/storage issues related to unfiltered sample matrix. Despite this, trends in
the concentration were discernible using the in situ data, although validation of
these trends requires more field work.
• The NH$_4$-N laboratory analysed data showed that concentrations were nearly
always below LOQ for the laboratory and as such were well below measurable
limits for the YSI sonde and electrode. This suggests that this analytical system is
not appropriate for this type of environmental setting. Despite this trends in NH$_4$-
N concentration were discernible from the sonde, although whether these are
analytical artefacts or genuine remains uncertain.
• Concentration of NOx-N were always higher than LOQ for both the in situ
NITRATAX sonde and the laboratory analysis. The two set of data show good
agreement, and exhibit similar classical NOx-N chemographs. However,
differences in the NOx-N are not linear and appear at lower concentration/higher
Q, with the in situ data giving lower concentrations than the laboratory measured
values. This may be an effect of the NITRATAX considering turbidity interference
at higher Q.

6. Summary
PHOSPHAX TP and NITRATAX NOx-N data show good agreement with laboratory data
in this environmental setting. However, PHOSPHAX TRP and YSI NH$_4$-N data were less
reliable as concentrations were below the instrumental limits. Both these instruments





generated data with repeatable trends in concentration, but trends that were not reflected in
the laboratory data which, in turn, was noisier. It is unclear whether the instrument trends
were genuine, or why they were not present in the laboratory data which is itself very
variable.

7.  Acknowledgements
The authors would like to thank the Stapledon Memorial Trust for the Travelling
Fellowship which enabled this work to be carried out. The work was also funded by the
Biotechnology and Biological Sciences Research Council (BBSRC) grants BB/J004308
(The North Wyke Farm Platform) and BB/P01268X/1 (Soil to Nutrition). The authors a
grateful to Jess Evans for her assistance with statistical support and to Verena Pfahler for
her help with the sampling.

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





| | | Sub-catchment 2 | | Sub-catchment 5 | | Sub-catchment 8 | |
|---|---|---|---|---|---|---|---|
| | | n | range | n | range | n | range |
| TP | In situ | 64 | 40 - 300 | 62 | 50 - 380 | 44 | 50 - 770 |
| µg P l⁻¹ | Lab | 30 | 46 - 365 | 31 | 68 - 428 | 38 | 48 - 706 |
| TRP | In situ | 65 | 20 - 70 | 63 | 0 - 60 | 45 | 0 - 60 |
| µg P l⁻¹ | Lab | 30 | 8 - 77 | 31 | 7 - 111 | 38 | 6 - 76 |
| NO$_X$-N | In situ | 130 | 0.62 – 1.7 | 126 | 1.6 – 4.8 | 119 | 0.72 – 2.1 |
| mg N l⁻¹ | Lab | 30 | 0.11 – 1.7 | 31 | 0.66 – 5.1 | 38 | 0.24 – 2.1 |
| NH$_4$-N | In situ | 130 | 0.12 – 0.32 | 126 | 0.04 – 0.14 | 119 | 0.55 – 1.5 |
| mg N l⁻¹ | Lab | 30 | 0 – 0.19 | 31 | 0.01 – 0.13 | 38 | 0.09 – 0.48 |


**Table 1.** Summary of values (min – max) measured in drainage from the three NWFP sub-

catchments using both the in situ automated analysers and laboratory analysis of manually

collected samples.



| | Catchment | Intercept | *p*-value | Slope | *p*-value | $R^2$ |
|---|---|---|---|---|---|---|
| | 2 | 22.40 | 0.033 | 0.79 | 0.002 | 0.85 |
| TP | 5 | -2.34 | **0.881** | 1.09 | **0.293** | 0.84 |
| | 8 | 20.45 | **0.259** | 1.04 | **0.482** | 0.86 |
| | 2 | 23.48 | 0.003 | 0.46 | 0.006 | 0.18 |
| TRP | 5 | 22.48 | 0.001 | 0.22 | 0.000 | 0.09 |
| | 8 | 19.08 | 0.003 | 0.41 | 0.000 | 0.21 |
| | 2 | 0.24 | 0.000 | -0.15 | 0.003 | 0.01 |
| NH$_4$-N | 5 | 0.05 | 0.000 | 0.67 | 0.001 | 0.67 |
| | 8 | 0.46 | 0.000 | 2.17 | 0.000 | 0.83 |
| | 2 | 0.63 | 0.000 | 0.54 | 0.000 | 0.98 |
| NOx-N | 5 | 1.20 | 0.000 | 0.70 | 0.000 | 0.96 |
| | 8 | 0.80 | 0.000 | 0.60 | 0.000 | 0.98 |


**Table 2:** Summary of linear regression outputs for in situ versus laboratory data. The

*p*-values that are bolded indicate intercepts or slopes that are not significantly different

to zero or one, respectively, at the 95% level.






| Parameter | Sub-catchment | ME | RMSE | NRMSE (%) |
|---|---|---|---|---|
| TP | 2 | -8.07 | 32.13 | 39.4 |
| | 5 | 12.05 | 43.47 | 50.1 |
| | 8 | 30.98 | 67.62 | 42.2 |
| TRP | 2 | 3.27 | 18.19 | 110.6 |
| | 5 | -12.68 | 26.13 | 117.1 |
| | 8 | -4.83 | 20.55 | 101.1 |
| NH$_4$-N | 2 | 0.20 | 0.21 | 638.7 |
| | 5 | 0.03 | 0.04 | 88.5 |
| | 8 | 0.82 | 0.84 | 626.6 |
| NOx-N | 2 | 0.30 | 0.40 | 68.3 |
| | 5 | 0.47 | 0.72 | 43.8 |
| | 8 | 0.47 | 0.54 | 80.1 |


**Table 3:** Summary of goodness of fit statistics for in situ versus laboratory data.



| Sub-catchment 2 | | | | Sub-catchment 5 | | | | Sub-catchment 8 | | | |
|---|---|---|---|---|---|---|---|---|---|---|---|
| TP ($\mu$g P l$^{-1}$) | | TRP ($\mu$g P l$^{-1}$) | | TP ($\mu$g P l$^{-1}$) | | TRP ($\mu$g P l$^{-1}$) | | TP ($\mu$g P l$^{-1}$) | | TRP ($\mu$g P l$^{-1}$) | |
| In situ | Lab | In situ | Lab | In situ | Lab | In situ | Lab | In situ | Lab | In situ | Lab |
| 50 | 45 | 20 | 36 | 50 | 80 | 20 | 42 | 50 | 48 | 20 | 7 |
| 60 | 77 | 20 | 40 | 70 | 72 | 20 | 36 | 160 | 161 | 20 | 27 |
| 170 | 174 | 60 | 61 | 210 | 197 | 50 | 83 | 390 | 371 | 50 | 61 |
| 140 | 138 | 50 | 52 | 150 | 133 | 40 | 47 | 250 | 194 | 60 | 56 |
| 70 | 73 | 20 | 25 | 60 | 68 | 20 | 46 | 70 | 60 | 30 | 15 |


**Table 4.** Comparison of TP and TRP data obtained from in situ analysers and laboratory
analysed manual samples of discharge sampled at the same time (i.e. genuine temporal pairs).





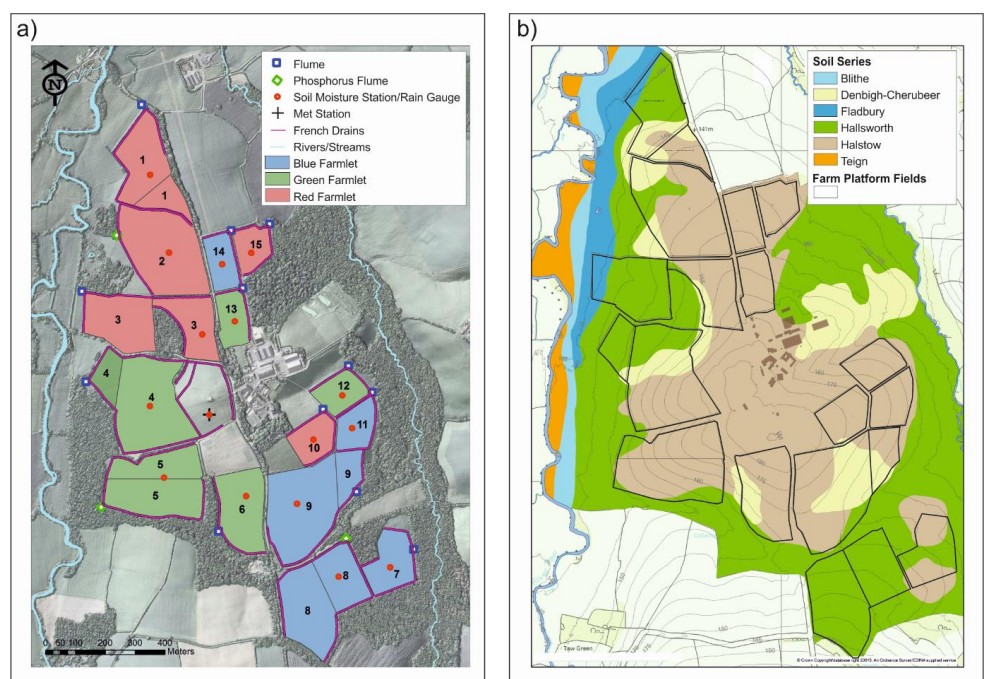

**Figure 1:** Maps of a) the North Wyke Farm Platform including infrastructure and b) soil series distribution.

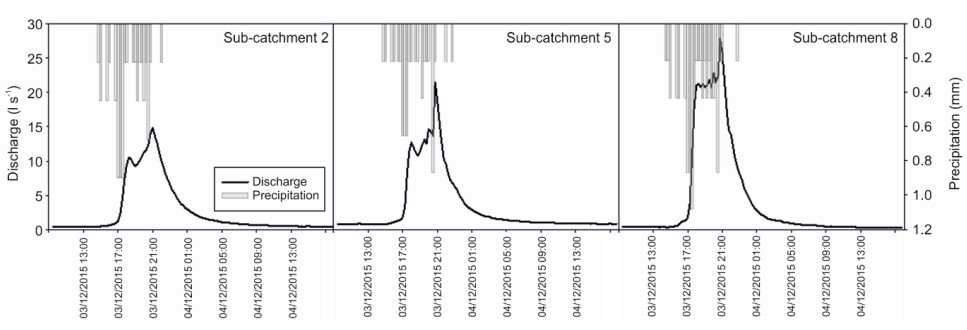

**Figure 2:** Discharge and precipitation for each sub-catchment.



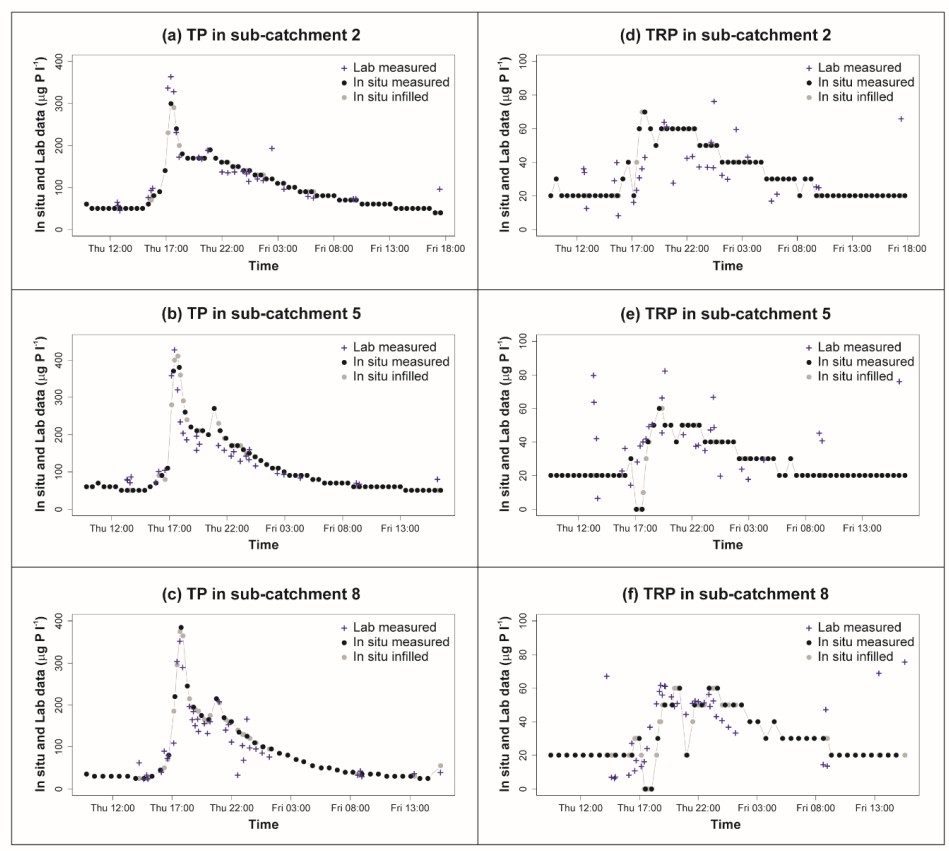


**Figure 3:** Time series plots for Total Phosphorus and Total Reactive Phosphorus showing the

data measured in situ relative to that measured in the laboratory physical sample, and the

modelled 'in filled' data.









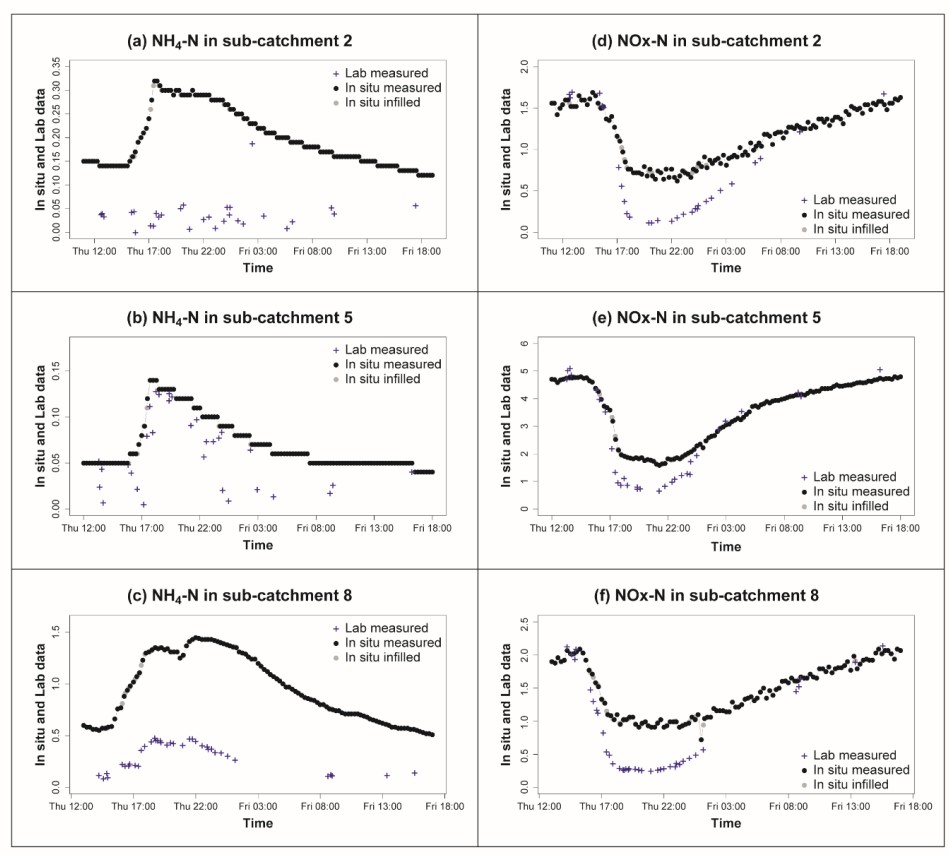


**Figure 4:** Time series plots for NH₄-N and NOx-N showing the data measured in situ relative

to that measured in the laboratory physical sample, and the modelled 'in filled' data.












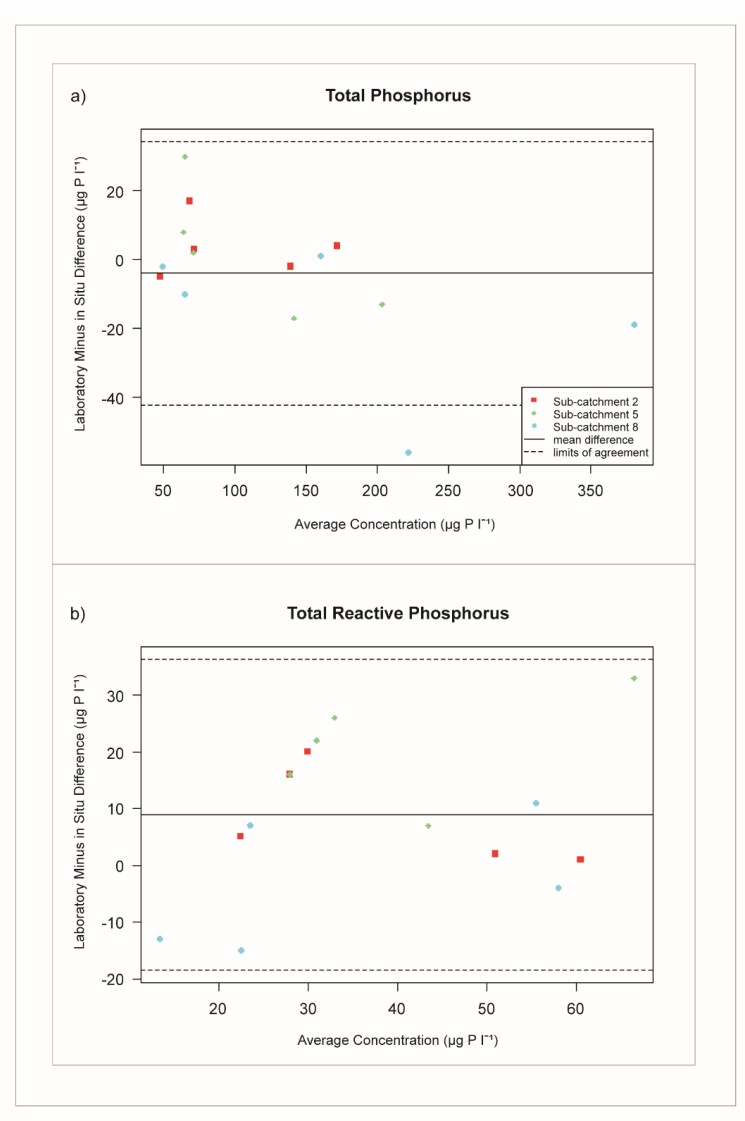


**Figure 5.** Tukey mean-difference plots showing the average concentration of the two

measurement methods against the difference between the two values (for TP and TRP only).








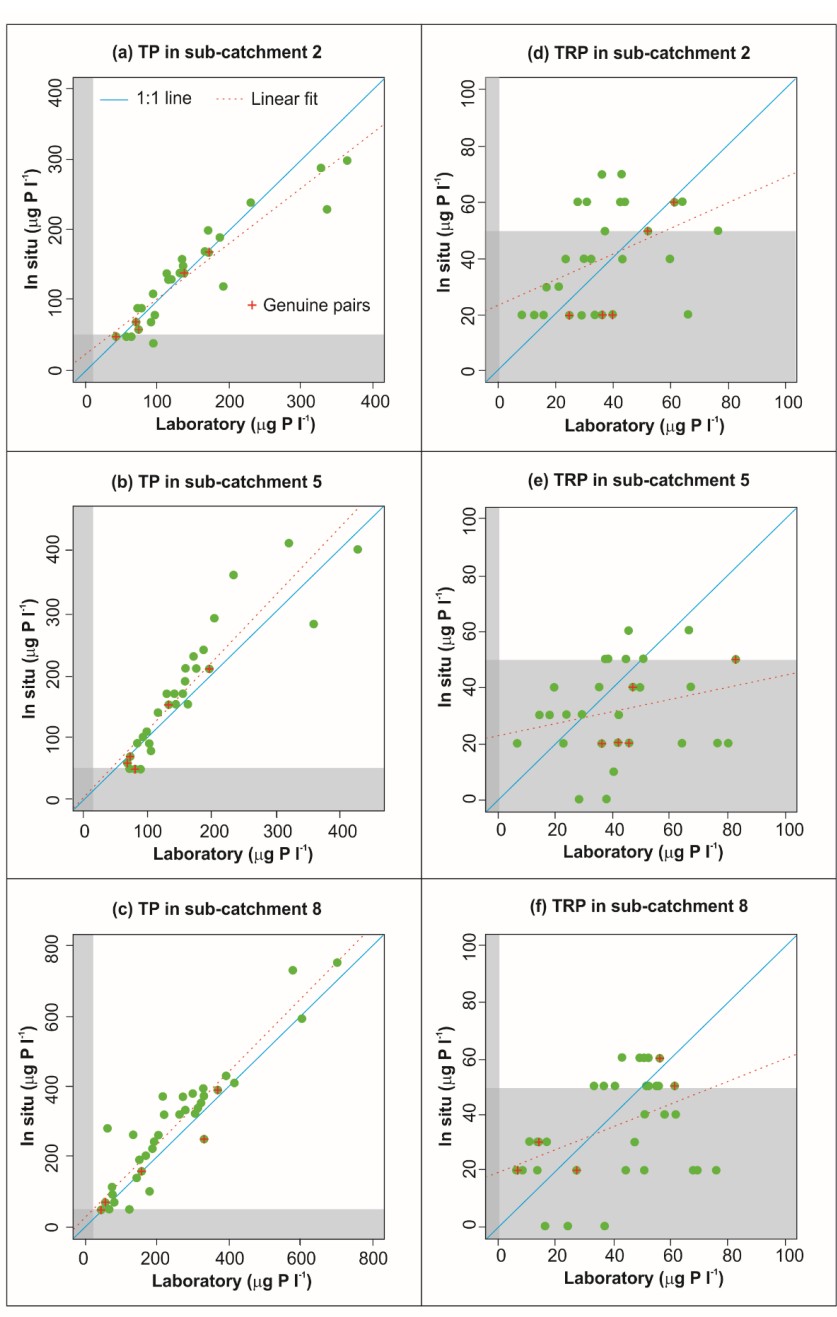


**Figure 6:** Scatterplots for paired TP and TRP data. Shaded grey areas indicate areas below

limits of analysis for accurate determination.



**Figure 7:** Scatterplots for paired NH₄-N and NOx-N data. Shaded grey areas indicate areas

below limits of analysis for accurate determination.