# Peer review of "Comparison of high frequency, in-situ water quality analysers and sensors with conventional"

_Hydrology and Earth System Sciences, 2017_

## Referee Comment (RC1) · Anonymous Referee #1 · 21 Feb 2018

General comments

This paper presents high-frequency nutrient data through a single storm event, produced by in situ analysers and sondes, and compares the P and N data to lab analysis of grab samples obtained using water samplers, at the North Wyke research facility in the UK. It assesses the relative merits of these particular auto-analysers for the monitoring of TP, TRP, NO3 and NH4. The paper is well-written and structured, and the figures are informative. However, I have some major issues that would need addressing before I could recommend publication.
Specific comments

1) The main focus of this study is to compare the autoanalyser data with the "gold standard" lab analysis. The degree of agreement between these two data sets is the key parameter that the authors use to evaluate the auto-analyser performance. However, there are a number of problems with this approach.

- The lab samples were only analysed after 48 hours (despite the text in the introduction discussing the issues of sample stability).

- The grab samples were not filtered for 48 hours, which is another major source of error.

- The nutrient concentrations at the study sites were often below the Limit of Quantification for the lab analysis

- The study sites were also below the LOQ for the autoanalysers and sondes.

Therefore, the lab data is somewhat unreliable, and the field instrument data is also unreliable. There is also an underlying assumption that the differences in the concentration data are due to inaccuracies in the autosamplers alone, which is unfair. It also ignores the previous studies (often using the Phosphax TP analyser) which have produced very good agreements with lab data (see referenced work by Palmer-Felgate, Rode, Skeffington, Mellander and Jordan) and others (Outram et al 2016, M Cohen). I would advise that the authors focus purely on the sites where both the lab and autoanalyser data is above the LOQ, to give a fair assessment of their effectiveness. Perhaps North Wyke was a poor choice for this particular study, but it does show that these instruments are maybe not suited to similar small agricultural catchments with low nutrient concentrations.

2) There is a major issue with the Phosphax data (Fig 3), which the authors state can only report data to the nearest 10 ug/l. I'm not sure this is correct, and other studies using the Phosphax (listed above) do not have the step changes that this study

presents. This needs to be addressed and the data corrected.

3) In terms of the nitrate time-series, I would suggest that the sonde data is much better than the lab data at low concentrations (Fig 4a and 4b).

4) There is a lack of detail in the autosampler methods. The samplers sampled at "predetermined flow thresholds". What were they and how were they chosen? What was the position of the sampling tube? Set depth? In main flow or marginal? How were the manual grab samples taken? Sample storage? How were they "kept cool"?

5) What lab quality controls were used? Were they external standards?

Technical corrections

Line 18: "this raises the question of whether high frequency lower precision data" This statement makes a presumption that the autoanalysers produce more inaccurate P and N data than standard grab sampling and analysis (which is subject to sample stability issues). This paper does not make the case that this is true

Line 26. The tone of this sentence is quite unfair to the instrument manufacturers and not supported by this study. It implies that the Phosphax and YSI data are poor, because they don't match with the lab data (which is also poor, due to storage issues and been below the LOQ). Change to "At low P and N concentrations, the PHOSPHAX, YSI and lab analyses were below LOQ". Better still, remove this data from the study.

Line 42. Change wet chemistry to colorimetry.

Line 67. "limited coverage of the stream cross section" should not be a problem in such small streams if sampling is taken from the main flow at mid depth.

Line 99. Remove green, red and blue farmlet. Not needed.

Line 105. Sentence doesn't make sense.

Line 142. Sentence doesn't make sense.

Line 242. Change to Nutrient response to discharge.

Line 266. Should it be Figure 4d-f?

Figure 1. I couldn't see the streams on the map.

Figure 3. Raw P data needs to be used, rather than being rounded to the nearest 10 ug/l. What happens to the TRP lab data at site 5 at 1pm Thursday? Is this real data, or a problem with the lab analysis? It really effects the plots in Figure 5. Remove the sites that are below LOQ.

Figure 4. YSI ammonium data seems to be much better than the lab data. Remove sites that are below LOQ.

Figure 5. The P data in Figure 3 goes up to 80 ug/l, but this data only goes up to ~65 ug/l. Is some data missing?

[Figure]

---

## Referee Comment (RC2) · Anonymous Referee #2 · 28 Feb 2018

This manuscript compares the results of high-frequency water quality monitoring with in-situ sensors or analyzers versus laboratory analyses. Parameters of interest are TP, TRP, NH4 and NOx. Only one runoff event is studied, in three different sub catchments of the North Wyke Farm Plateform (UK).

Although the issue of measurement quality deserves an article in HESS, there are several flaws in the approach, that would need to be addressed before the manuscript can be accepted:

[Figure]

1. Lack of a "good" reference measurement. Lab analysis is assumed to be the true value although storage times of (unfiltered) samples was 48h. The NH4 graph seems to indicate problems of storage, as does the largest "noise" visible for other parameters in the lab analysis. In-situ equipment, however, can be subjected to poor resolution of the recorded value (10 $\mu$gP/l) which is interesting to highlight.

2. It would be interesting to present the method used to determine measurement accuracy and LOQ for both in situ and lab equipment, and make sure that they are the same. In the current version of the manuscript it seems that the authors quantified the LOQ of their lab equipment but used the LOQ provided by the manufacturer for their in situ equipment. And study the effect of sample storage on measurement accuracy and LOQ, in addition to the lab analysis alone. Some data analyses could be performed with the data > LOQ for both lab and field procedure.

3. One possibility for a fair comparison would be to bring samples of known concentrations (prepared in the lab and immediately analyzed in the lab) to the field equipment. By doing this, there would be no issue of sample storage and both methods could be compared. Another factor that make the comparison unfair is that the lab equipment is calibrated (probably?) and not the field equipment (maybe the PHOSPHAX is).

4. The comparison would be more solid with more storm events monitored, as only 30 data points are currently available (of which 5 are genuine paired samples), which seems to be little for some of statistical analyses done.

Detailed comments

Line 16 "The downside to this approach is that the data can be subject to more 'noise'" this seems to be in contraction with the data collected in this study, where lab analyses appear to be more subjected to analytical "noise". In-situ equipment, however, can be subjected to poor resolution of the recorded value (eg 10 $\mu$gP/l) which is interesting to highlight.

Line 18 and line 81 "This raises the question of whether high frequency, lower precision data are better than low frequency, higher precision data." I think that the assumption that sample collection, storage and analysis has a higher precision than in situ equipment should be tested, including the effect of storage. It would be interesting to clarify "better" here, because depending on the objective of the monitoring (assess mean concentration or load during a given period, look at the temporal dynamics, et.c) either of the two methods can be better.

Line 25 "the NITRATAX can under report the concentration" or the lab measurement overestimates it.

Line 58 "In situ devices remove sample storage requirements and provide a means of avoiding water sample storage-associated chemical transformations" this is important and should be considered in the manuscript by analyzing the impact of storage on the quality of measurement.

Line 106 the sub catchments are not visible in Figure 1b

Line 166 "quality controls" please give more details, including the number of samples used.

Line 187 "on one of three occasion" that is to say 10 out of 30?

Line 204 "they should lie on the 45° line" call it the 1:1 line?

Line 218 "A final, but limited analysis was also conducted on the genuine paired samples found for TP and TRP only - i.e. only five pairs for each sub-catchment" I doubt this is a sufficient number for the statistical analysis conducted".

Line 247 "In all three sub-catchments, TP data from both in situ analysers and the laboratory analysed grab samples exhibited a positive relationship with discharge (Figure 3a-c)." I did not see discharge in Figure 3.

Line 251 "In all cases, the chemographs generated by both analytical approaches appear similar and match the responses reported elsewhere" line 253 "Such relationships with discharge are less clear" to vague, use metrics and statistics

line 255 "and possibly even a two peaked chemograph" I did not see it.

Line 272 "Where high concentrations of NH4-N occur as spikes associated with discharge, it is often more related to incidental losses of recently applied NH4- N as a result of farmland management practices" interpretation should be moved to discussion section.

Line 257 "by very high $R^2$" used a fixed threshold throughout the manuscript to consider that $R^2$ is "high" and write $R^2 > ...$ instead

Line 395 "indicating that the in situ data were consistently lower than that measured in the laboratory." Not really consistently lower in Figure 3.

Avoid subjective comments such as "albeit not one that is ideal" line 352, use metrics and statistics instead. Avoid double negative such as Line 269 "is not unusual"

Table 1: add other statistics such as the mean and standard deviation

Table 2: add units

---

## Referee Comment (RC3) · Anonymous Referee #3 · 19 Mar 2018

General comments: The manuscript of Granger et al. aims at comparing in-situ high frequency water quality analysers and sensors with conventional water sample collection and laboratory analyses, focusing on phosphorus and nitrogen species. Although the topic offers potential for an in deep analysis of the precision of high frequency water quality measurements the present paper offers only a very narrow view of the topic with regard to investigated compounds, range and amount of analysed data and analytical devices. The main concerns I have are as follows: Regarding NOx-N analysis only the Nitratax UV sensor is included in the study. Although the tested sensor could

provide comparable data to the measured using samples analysed in the laboratory, this sensor is one with the lowest precision of devices currently on the market (0.5 mg NOx-N L-1). Other sensors have much higher precision. This fact is not discussed in the paper and may result in a misleading assessment of optical nitrate sensors. A comprehensive analysis of most important UV sensors has already been given by the USGS some years ago (https://pubs.usgs.gov/tm/01/d5/) (2013). This study is not cited in the manuscript. The abstract suggests that always a choice has to be made between high frequency measurements of low precision and low frequency measurements of high precision. This is not proofed by the results. I cannot follow this arguing because at least in case of nitrate optical high frequency measurements do not show inevitably lower precision than laboratory measurements if the sensors are accurately maintained. Studies using high frequency in situ data mostly validate the sensors with laboratory data. This is true for optical sensors, see e.g. Pellerin et al. 2015, Heffernan and Cohen 2010 as well as for Phosphax analysers (e.g. Halliday et al. 2014). Therefore a rigorous analysis of the available literature on the precision of high frequency measurement devices would likely give a comprehensive picture on the measurement uncertainties related to these analysers. This is not the case for the presented study. A comprehensive analysis of already available studies and a discussion on how the given results fit into this picture is missing. The amount of data presented in the manuscript is small. Only one runoff event of each catchment has been included in the analyses. Because all samples were taken on the same dates the range of data is narrow and very similar for all three catchments. For example all SRP values of the three catchments range between 0-70 $\mu$g P L-1. Also NOx-N ranges are similar within the three different catchments during the analysed event (below 5.1 mg N L-1). This limits possible conclusions from this investigation. Furthermore a discussion on other P analysers, e.g. Cycle-PO4 (Wetlabs), Cohen et al. (2013), which may provide different results, is missing. It is well known that other compounds like e.g. turbidity may interfere with nitrate sensor measurements and may affect the precision of measurement. But the presented study does not include further compounds, even not in the discussion. This

strongly limits interpretation of the results. Another important limitation of the NH4-N analysis is given by the fact that the in situ analyser has been used for data ranges which were well below the detection limit of the instrument. Therefore I cannot understand why this analysis has been carried out at all. Because of all these limitations I do not see that the manuscript offers important outcomes beyond what already is known and therefore I cannot support the publication of the manuscript in HESS.

Literature:

Brian A. Pellerin, Brian A. Bergamaschi, Bryan D. Downing, John Franco Saraceno, Jessica D. Garrett, and Lisa D. Olsen Optical Techniques for the Determination of Nitrate in Environmental Waters: Guidelines for Instrument Selection, Operation, Deployment, Maintenance, Quality Assurance, and Data Reporting, Chapter 5 of Section D, Water Quality Book 1, Collection of Water Data by Direct Measurement, Techniques and Methods 1–D5, USGS, 2013

Cohen, M. J., M. J. Kurz, J. B. Heffernan, J. B. Martin, R. L. Douglass, C. R. Foster, and R. G. Thomas. 2013. Diel phosphorus variation and the stoichiometry of ecosystem metabolism in a large spring-fed river. Ecological Monographs 83:155-176.

Heffernan, J. B.; Cohen, M. J.; Frazer, T. K.; Thomas, R. G.; Rayfield, T. J.; Gulley, J.; Martin, J. B.; Delfino, J. J.; Graham, W. D., Hydrologic and biotic influences on nitrate removal in a subtropical spring-fed river. Limnol. Oceanogr. 2010, 55, (1), 249-263.

Halliday, S., Skeffington, R., Bowes, M., Gozzard, E., Newman, J., Loewenthal, M., et al., 2014. The water quality of the River Enborne, UK: observations from highfrequency monitoring in a rural, lowland river system. Water 6, 150–180.

Pellerin, B. A., B. A. Bergamaschi, R. J. Gilliom, C. G. Crawford, J. Saraceno, C. P. Frederick, B. D. Downing, and J. C. Murphy. 2014. Mississippi River Nitrate Loads from High Frequency Sensor Measurements and Regression-Based Load Estimation. Environmental Science & Technology 48:12612-12619

---

## Author Comment (AC1) · 3 Apr 2018

Re: Specific comment 1) We obviously agree and acknowledge the issues of the comparison between approached and the LOQ problem. Obviously any analysis/sampling/approach introduces error however, we are as confident as we can be about the quality of the lab data. The error introduced between 'sampling' and 'analysis' however could perhaps be emphasised better which we will undertake in the revision process. As the reviewer points out, perhaps small agricultural catchments are not best suited for this equipment, which is an aspect of the work that is worth reporting

and noting.

Re Specific comment 2) The data reported by the Phosphax IS rounded to the nearest 10 ug/l at our site. This may have been a function of the telemetry system that transmits the data back to the main site. It is something we can report on in the text when we have clarified why this is the case

Re Specific comment 3) This maybe the case but the only QC'd data we have is from the lab. Therefore we compare to it as the main point of reference. There was a divergence of concentration between the two approaches at low NO3 concentrations but we were unable to clarify why that occurred. My feeling was that it was a sonde issue of some description, however we cannot confirm either way and it would be wrong to side with one approach or the other on a 'feeling'.

Re Specific comment 4) We can present information on the 'predetermined flow thresholds' if required, however we did not feel as though that detail would have helped the reader in the understanding of the results/discussion. For brevity we didn't include other than to say samples were taken throughout the flow event. Similarly with sampling position: as this is not a river channel, but instead a small circular concrete culvert with limited space for sample variation within the flow we did not include the information for brevity. However this can all be provided should it be felt necessary. More detail on the manual grab sample collection and storage will be provided.

Re Specific comment 5) Given our position that we compare autoanalyser data to the lab data because it was analysed in a QC'd environment we will provide information on the analytical quality process in out laboratory.

---

## Author Comment (AC2) · 16 Apr 2018

Re Specific comment: 1) Information on storage/filtering effects can be brought into the discussion in the revised manuscript. It is important to note however that this was a 'real' comparison and was meant to simulate what actually happens in reality. There are always issues in data quality with whatever approach is taken to collect information on water quality parameters.

The issue of 10 ugP/l resolution was also flagged by Reviewer 1 and it is something

we will report on in the revised manuscript.

Re Specific comment 2) This is something that could be done in future studies; however, it was not the aim of this study, which was to compare the two approaches in a realistic field setting i.e. manual sampling/lab analysis v in situ automated equipment. To address this comment, we will flag up the possibility of doing further work in the future.

It would be very difficult to compare LOQ on the two differing approaches in the way this reviewer implies as we feel that as their modes of operation are different, a standardised assessment of LOQ would in most cases not be possible. End users of the in situ equipment are all using the LOQ provided by the manufacturer, so again, our experiment captures what is happening in reality.

As with point 1, additional comment/discussion on sample storage in relation to LOQ will be included in the revised manuscript. However it is worth noting that in reality, when such studies are undertaken, all 'manual' samples are subject to this issue, and it is not a consistent issue, as by the time samples are presented to the laboratory, invariably some samples will be older than others just by the nature of the temporal sampling.

Re Specific comment 3) Similarly to point 2, this is something that could have been done differently, but the comparison was a 'real' field test study. The nature of whether field based analyses are calibrated is one of the important differences in the use of such equipment. Field based monitors ARE regularly serviced and calibrated, some (i.e. Phosphax) even run occasional in situ checks (this information can be included/made clearer in the revised manuscript). But this is completely different to laboratory analyses which are calibrated (over a ranged of values) and QC'd every time the lab bench equipment is used. It is one of the trade-offs made when considering whether to go for data rich in situ monitors, or manual/auto-sampler samples which are more expensive, time consuming, labour intensive, and time delayed, but yield more reliable analytically

data. This is why we chose to compare field in situ kit TO the laboratory data and not vice versa – i.e. we are wanting to test what actually happens in reality with end users. We will acknowledge in the revised paper that both approaches have their weaknesses; a challenge faced by all researchers in this field.

Re Specific comment 4) More data is always better. However, we don't believe that our analysis of one storm event at three sites is invalid. The statistical analysis reported in the paper has been provided and validated by a statistician. Again, the reporting of only 5 genuine pairs is the reality of this approach and the challenge faced by field researchers. Manually collected genuine sample pairs will always be much lower in number when compared to in situ equipment, which in turn rarely correspond temporally to auto-sampler samples. We have tried to reflect this trade-off in the manuscript. It is why we specifically included the 5 genuine sample pairs as part of the discussion in the draft paper.

Re detailed comments. Many of the more 'detailed' comments are similar to those raised in the general comments of Reviewer 1 and we will address them accordingly when we address Reviewer 1 in the revised paper

---

## Author Comment (AC3) · 20 Apr 2018

1) Reviewer 3 states that "the present paper offers only a very narrow view of the topic about investigated compounds, range and amount of analysed data and analytical devices" and then goes on to dismiss the Nitratax UV as "one with the lowest precision of devices currently on the market", furthermore, "a discussion on other P analysers" is missing. However, their point and the general context of much of Review 3s criticism, is that more sensors should have either been used in the study, and/or a more in-depth review of available sensors should have been undertaken. We feel this reviewer

misses the point of our study, which was not to review all the sensors available for every compound and circumstance but simply to compare, in a real setting, the sensors at our disposal with manually collected samples subject to standard laboratory analysis. Therefore, it is not our intention to give "a misleading assessment of optical nitrate sensors", indeed we are currently using one so we are in no way dismissing them as a useful tool in scientific environmental studies. In our revised manuscript, we can direct readers to the reports and other assessments (e.g. the USGS work) suggested by Reviewer 3 within the introduction and expand upon the evidence for and against the trade-offs between high frequency measurements of low precision and low frequency measurements of high precision. We will also emphasize to readers that our study is in no way dismissing the use of sensors.

2) Reviewer 3 strongly defends the accuracy and validity of in situ sensors as "Studies using high frequency in situ data mostly validate the sensors with laboratory data. This is true for optical sensors, see e.g. Pellerin et al. 2015, Heffernan and Cohen 2010 as well as for Phosphax analysers (e.g. Halliday et al. 2014)." However, Reviewer 3 also indicates that "It is well known that other compounds like e.g. turbidity may interfere with nitrate sensor measurements and may affect the precision of measurement." It is this degree of uncertainty between in situ and laboratory data, and the confounding features and trade-offs of both we sought to examine in our study. Indeed Reviewer 3 also states that we do "not include further compounds, even not in the discussion". This is untrue as we do make mention of other compounds affecting the ISEs potentially leading to some of the trends we see although we could not confirm that it was the case. However, we will expand this issue in the discussion within a revised version of the manuscript and this expansion will place our results in the context of the findings of previous work comparing in situ sensors and laboratory analyses.

3) Reviewer 3, criticises the "amount of data presented in the manuscript is small. Only one runoff event of each catchment has been included in the analyses. Because all samples were taken on the same dates the range of data is narrow and very similar

for all three catchments". This point is indeed a valid one. However, these instruments ARE being used in this environment, including NH4-N sensors, and therefore it is valid to examine them in this context, and to assess whether they are suitable. Such as assessment can be expanded upon in the revised manuscript to ensure we highlight potential limitations associated with the fact we only sampled a single storm. Likewise, expanded discussion can also include consideration of additional sensors including the one cited in the review comment here.